# A Combined Experimental and Theoretical Study of Nitrofuran Antibiotics: Crystal Structures, DFT Computations, Sublimation and Solution Thermodynamics

**DOI:** 10.3390/molecules26113444

**Published:** 2021-06-05

**Authors:** Alex N. Manin, Ksenia V. Drozd, Alexander P. Voronin, Andrei V. Churakov, German L. Perlovich

**Affiliations:** 1G.A. Krestov Institute of Solution Chemistry of the Russian Academy of Sciences, 1 Akademicheskaya St., 153045 Ivanovo, Russia; anm@isc-ras.ru (A.N.M.); ksdrozd@yandex.ru (K.V.D.); Flox-av@yandex.ru (A.P.V.); 2Institute of General and Inorganic Chemistry, Russian Academy of Sciences, 31 Leninsky Prosp., 119991 Moscow, Russia; churakov@igic.ras.ru

**Keywords:** nitrofurantoin, furazolidone, sublimation thermodynamics, transpiration method, crystal structure, crystal lattice energy, solubility

## Abstract

Single crystal of furazolidone (FZL) has been successfully obtained, and its crystal structure has been determined. Common and distinctive features of furazolidone and nitrofurantoin (NFT) crystal packing have been discussed. Combined use of QTAIMC and Hirshfeld surface analysis allowed characterizing the non-covalent interactions in both crystals. Thermophysical characteristics and decomposition of NFT and FZL have been studied by differential scanning calorimetry (DSC), thermogravimetric analysis (TG) and mass-spectrometry. The saturated vapor pressures of the compounds have been measured using the transpiration method, and the standard thermodynamic functions of sublimation were calculated. It was revealed that the sublimation enthalpy and Gibbs energy of NFT are both higher than those for FZL, but a gain in the crystal lattice energy of NFT is leveled by an entropy increase. The solubility processes of the studied compounds in buffer solutions with pH 2.0, 7.4 and in 1-octanol was investigated at four temperatures from 298.15 to 313.15 K by the saturation shake-flask method. The thermodynamic functions of the dissolution and solvation processes of the studied compounds have been calculated based on the experimental data. Due to the fact that NFT is unstable in buffer solutions and undergoes a solution-mediated transformation from an anhydrate form to monohydrate in the solid state, the thermophysical characteristics and dissolution thermodynamics of the monohydrate were also investigated. It was demonstrated that a combination of experimental and theoretical methods allows performing an in-depth study of the relationships between the molecular and crystal structure and pharmaceutically relevant properties of nitrofuran antibiotics.

## 1. Introduction

Nitrofurans (NFs) belong to the class of active pharmaceutical ingredients (APIs) active against both Gram-positive and Gram-negative microorganisms, and are routinely used to treat urinary infections [1]. They are also reported to possess anti-protozoal and antihelmintic activity. Two members of this class, nitrofurantoin (NFT) and furazolidone (FZL) (Figure 1) have been synthesized and approved for clinical use in the 20th century, but are still actively used nowadays [2]. The WHO includes NFT in its Model List of Essential Medicines [3]. In the era of antibiotic resistance, NFs draw high attention due to antimicrobial resistance to more common APIs [4,5]. A number of new potent NFs with antibacterial, antiprotozoal, and antitubercular activity (including multidrug-resistant infections) have been appeared in the last decades [6]. At the same time, old APIs are being repurposed as promising antitumor drugs [7,8].

Most NFs belong to class IV of the Biopharmaceutical Classification System and, as a consequence, exhibit low oral bioavailability, caused by poor aqueous solubility and permeability. As follows, an effective treatment with this class of APIs requires the use of high doses, which often leads to severe side effects [9]. However, the origins of poor aqueous solubility of nitrofurans have not been yet studied intensively.

The development of new drugs requires an understanding of the mechanisms of the drug transport and drug delivery processes to biological targets. These processes include the stages of drug release (dissolution), absorption, distribution, and passive transport and are determined by the solvation characteristics of the drug molecules [10]. Experimental determination of compounds solvation characteristics is a non-trivial task, since one needs to investigate the two independent processes: sublimation and dissolution. There are almost no such works in the literature, so we tried to analyze the solvation contributions based on the experimental data obtained for the selected compounds.

Measurement of the sublimation thermodynamic functions for many drug substances is difficult due to their extremely low volatility and thermal instability. However, in recent years, works whose authors are trying to fill the gaps by various experimental and theoretical methods have gradually begun to appear [11,12,13].

At the same time, the crystallographic and thermodynamic studies of the identical compounds are usually carried out by independent scientific groups. As a result, the influence of the molecular packing in crystal on the energy characteristics of the dissolution and sublimation processes is disregarded, which is important, in particular, for studying of API polymorphs and pharmaceutical cocrystals [14,15,16].

In the present paper, the thermodynamics of the sublimation and dissolution processes of two related nitrofuran antibiotics, nitrofurantoin (NFT) and furazolidone (FZL), was investigated. The relationship between the molecular and crystal structure was disclosed. In particular, the crystal structure of FZL was determined for the first time. Experimental data on the temperature dependences of the saturated vapor pressure of NFT and FZL were obtained. The influence of the presence of a hydrogen bond donor and molecular packing in a crystal on the thermodynamic functions of the sublimation, dissolution and solvation processes was studied by the experimental and computational approaches. The dissolution studies were carried out in three pharmaceutically relevant media: isotonic aqueous buffer solutions with pH 2.0 and 7.4, and in 1-octanol. The solvents for this study were chosen taking in account their ability to simulate the biological media of an organism. Buffer solutions with pH 2.0 and 7.4 imitate the pH of the acidic medium of the gastric fluid and the blood, respectively. 1-Octanol models the properties of phospholipids of biological membranes, therefore, the obtained solvation characteristics in 1-octanol can be very useful for predicting the processes of passive diffusion (transport) through the cells and the distribution of the drug molecules at the boundaries of the biological membranes, which is also in demand during the development and formulation of poorly permeable APIs.

## 2. Results and Discussion

### 2.1. Crystal Structure Analysis

NFT has two known anhydrous polymorphs (*α*- and *β*-) with Cambridge structural database (CSD) codes of LABJON01 for *α*-form and LABJON02, LABJON03 and AZAXEG for *β-*form, respectively, and two monohydrated forms (I and II) [17,18,19,20]. The *β*-form of NFT is the solid form stable at room temperature. To date, the crystal structure of FZL has not been determined yet.

Therefore, the search for the optimal conditions for the preparation of the FZL single crystal in order to determine its crystal structure was one of the objectives of this work. Knowledge of the specific features of molecular packing in the crystal lattice will make it possible to reveal the structure–property relationship for the compounds under study. As a result of varying a number of conditions, including the choice of organic solvent and crystallization method, the orange plate-shaped crystals of FZL were obtained by slow crystallization from acetone. Crystallographic data of FZL are given in Table 1.

The ORTEP diagram of a FZL molecule in the crystal along with atom numbering scheme are presented in Figure 2a. Similar to other nitrofuranes, FZL adopts a planar conformation in crystal, which facilitates the crystal packing with distinguishable layers and columns (Figure 2b,c).

The NFT and FZL molecules are structurally similar, which is partially reflected by the similarity in their molecular packing. In particular, both crystals have a common hydrogen-bonded chain packing motif displayed in Figure 2b and Appendix A. QTAIMC analysis has revealed that the total energy of non-covalent interactions in the chain motif formed by three C−H···O hydrogen bonds is comparable in both crystals and is ~30 kJ·mol^−1^ (Table 2). The CSD survey of NFT-containing structures has found an identical chain motif in NFT methanol solvate (ULECAQ), NFT picoline solvate monohydrate (NESKED) and in NFT cocrystal with 4-hydroxybenzoic acid (LIGYOR), which signalizes of its high stability.

In contrast to the FZL crystal, in which the hydrogen-bonded chains are held within the 2D layer only by weak interactions, the chains in stable *β*-form of NFT are linked via hydrogen bonds of different strength, displayed in Appendix A. This is reflected in increasing the total bond energy (Table 2). The N1−H1···O2 hydrogen bond in the NFT crystal belongs to a moderate type with a large electrostatic component (*ρ_b_* = 0.029 a.u., ∇^2^*ρ_b_* = 0.108 a.u.) and has the interaction energy of 27.9 kJ·mol^−1^. Due to this, the total interaction energy between the adjacent chains inside the 2D layer in *β*-form of NFT is stronger than that within the chain (Table 2). Additional intermolecular contacts in the NFT crystal are formed by the N−H group and the second C=O fragment, which are absent in the FZL molecule, which leads to more than double increase in the packing energy of chains within a layer compared to the FZL crystal (40.7 against 18.6 kJ·mol^−1^).

The FZL planes formed by the weak hydrogen bonds are held together via π∙∙∙π stacking interactions. Figure 2c shows that the FZL packing is characterized by the formation of layers, while the arrangement of the hydrogen-bond chains of NFT molecules is of the slipped stacking type (Appendix A) [17,21]. From the analysis of the contributions to the *E*_latt_(0K) value it can be concluded that the π∙∙∙π stacking interactions between the layers make a significant contribution to the packing of molecules in both crystals, which is comparable to the hydrogen bonding energy (Table 2). Since the FZL molecules in the crystal have a perfectly planar conformation and are located on the mirror planes, the packing energy of the layers in the crystal is 10 kJ·mol^−1^ higher than in the NFT crystal (74.8 kJ·mol^−1^ for FZL against 65.2 kJ·mol^−1^ for NFT). Two pairs of C−H···O bonds formed by symmetrically located C2−H2 and C3−H3 groups and oxygen atoms of the molecules located in the adjacent layers also play an important role in the stabilization of the FZL layers. The layers in the NFT crystal are parallel to the (−1 0 2) plane and are not absolutely flat due to the non-planar conformation of the NFT molecules (Appendix A).

Additional information about the molecular packing architecture can be obtained from the Hirshfeld surface analysis (Figure 3). According to it, nearly half of the contact surface in both crystals is occupied by H···O and H···N contacts representing the hydrogen bonds. This result agrees well with the fraction of hydrogen bond energies in the *E*_latt_ value derived from QTAIMC. In NFT, the relative area of H···O contacts is 2% higher, from which one can state that the C−H···O interactions in FZL only partially compensate for the absence of a stronger N−H···O bond. The second largest contribution comes from H···H contacts, which are responsible for non-specific dispersive interactions. This value is 50% higher in the FZL crystal, which correlates with higher number of hydrogen atoms in the molecule and non-zero fraction of H···H interactions observed by QTAIMC. Surprisingly, the H···C contacts occupy 7 to 9% of the Hirshfeld surface. This type of contacts are commonly associated with intermolecular C−H···π interactions; however, neither metric criteria nor QTAIMC analysis have not found a single C−H···π contact in crystal structures consisting of parallel layers. Therefore, we have to consider such unusually high fraction of H···C contacts as an artifact of the calculation algorithm. Other contacts between the heavy atoms on the 2D fingerprint plot were classified as signs of the stacking interactions, which can be seen from their location on the upper and lower parts of the molecular surface. The most abundant of these contacts are C···O (7.5–9.5%), O···O (4.5–6.1%) and N···O (3.5–5.1%). Taken together, they occupy about 28% in the FZD crystal and 24% in the NFT crystal, which only partially consists with the fractions of π···π stacking interactions in the lattice energy according to QTAIMC.

### 2.2. Thermal Study of the Compounds

Thermal analysis coupled with differential scanning calorimetry, thermogravimetry and mass-spectrometry was applied to study the thermophysical properties of NFT and FZL. The DSC/TG curves for both compounds are shown in Figure 4.

According to the DSC/TG curves obtained, NFT and FZL are stable up to the start of the sample melting. In both cases, some overlapped endothermic and exothermic events are observed in the range of 530–575 K for NFT and 520–550 K for FZL, that are related to the melting of API (endo-effect) and its degradation (exo-effect). Therefore, it is not possible to calculate the enthalpy of fusion of the compounds. For NFT, a single endotherm at $T_{fus}^{onset}$ = 541.2 ± 0.2 K ($T_{fus}^{peak}$ = 543.0 ± 0.2 K) is found to correspond to the API melting. The obtained value of the NFT melting point is in good agreement with the literature data [15,22]. As mentioned above, two polymorphs of NFT (*α*- and *β*-) are known, the DSC curves of which are similar without any phase transitions, indicating the transformation of one polymorphic form into another. Moreover, the peak melting temperatures of NFT polymorphs are very close (548 K for *α*-form and 543 K for *β*-form) [22]. Therefore, it is recommended to take an additional PXRD for more accurate identification of the polymorphic form of NFT (Appendix A). The melting point of FZL is $T_{fus}^{onset}$= 527.9 ± 0.2 K ($T_{fus}^{peak}$= 529.4 ± 0.2 K). Analysis of the FZL melting temperature according to the literature showed a good agreement with the results of the present work [23,24]. According to the TG curves, the melting and degradation processes for both compounds are accompanied by the one-stage mass loss: ~52% for NFT and ~55% for FZL. In addition, the more detailed analysis of the obtained thermograms showed that the mass loss of the compounds begins at a temperature of 2–5 K below the melting temperature of APIs. It may indicate a potential sublimation of the compounds.

Besides the DSC/TG experiments, the stability of NFT and FZL in the gas phase prior to the sublimation experiment was investigated by mass-spectrometry. The mass-spectra for both nitrofurans were obtained at various temperatures: both before (which was especially necessary for sublimation experiment) and after the melting of API (Appendix A). The mass-spectrometer analysis showed the absence of NFT or FZL in gas phase in the temperature interval under study. The analysis of the obtained mass-spectra for both compounds, in particular, the monitoring of the changes in the relative intensity of the peaks with m/z, made it possible to conclude that the NFT and FZL degradation products in gas phase appear only at the temperature above the melting point of APIs. Therefore, in both cases, it is possible to identify two key decomposition products for each of the APIs, the relative intensity of the peaks of which increases with the increasing of the temperature as compared to the other peaks. NFT and FZL are structurally related compounds and, as follows, their fragmentation as a result of the thermal degradation in a gas phase has a similar path, namely, a cleavage of the single N−N bond between the two rings leads to the formation of 1-(5-Nitro-2-furyl)methanimine (m/z = 139) and hydantoin (m/z = 100) for NFT or oxazolidinone (m/z = 87) for FZL. Thus, we have proven that there are no degradation products in a gas phase at the temperature below the melting point of NFT or FZL. Therefore, it is possible to measure the saturated vapor pressures of APIs by the transpiration method, because the experiment is carried out at temperatures well below the melting points of the compounds.

### 2.3. Vapor Pressure and Sublimation Thermodynamics

The transpiration method was used to measure the vapor pressure of NFT and FZL. The experimental temperature dependences of NFT and FZL saturated vapor pressures are summarized in Appendix A and graphically shown in Figure 5. The study of NFs was carried out over a broad temperature range 423.1–467.4 K, corresponding to a pressure range from 0.004 to 0.122 Pa. The calculated standard thermodynamic functions of the sublimation are summarized in Table 3. It is worth noting that the experimental sublimation enthalpies of both compounds agree well with the theoretical lattice energy computed according to Equation (13): 146.5 ± 0.7 vs. 142.1 kJ·mol^−1^ for NFT and 127.9 ± 1.1 vs. 124.4 kJ·mol^−1^ for FZL. This proves that the additive scheme based on the QTAIMC descriptor *G_b_* yields accurate *E*_latt_ values for molecular organic crystals formed mainly by moderate and weak hydrogen bonds as stated in our earlier publications [13]. At the same time, the heat capacities and entropies obtained from the vibrational frequencies calculations for isolated molecules and bulk crystals were found to deviate significantly from the experimental values, leading to incorrect $\Delta_{cr}^{g}G_{m}^{0}(298.15K)$ values (Appendix A).

In our previous work [13], we proved that the transpiration method makes it possible to study the sublimation processes of thermally unstable compounds. We have shown previously by mass-spectrometry that NFT and FZL are chemically stable compounds at the temperature below the melting point. During the sublimation experiments, the absence of the phase transitions in the whole temperature range was shown by the comparison of the electronic absorption spectra of the sublimated substance with the UV spectrum of original samples before the experiment. Whenever the UV absorption spectra of the sublimated samples of NFT and FZL coincided with the spectra of the original samples (Appendix A), it was proved that NFT and FZL remained stable in the gas phase over the entire temperature range. The chemical stability of NFT and FZL in solid state was confirmed by the powder X-ray diffraction analysis of the samples tested after the sublimation experiment. The NFT and FZL diffractograms before and after the experiment were identical (Appendix A).

Based on the data presented in Table 3, the saturated vapor pressure of NFT is two orders of magnitude lower than of FZL. The sublimation Gibbs energy of NFT is higher, than that of FZL by more than 10 kJ·mol^−1^. The superiority of the NFT sublimation enthalpy value over FZL is comparable to the value of the energy of one hydrogen bond (~20 kJ·mol^−1^). On the other hand, the equality of the ratio of the enthalpy and entropy terms for the compounds under study indicates that the gain in the crystal lattice energy is leveled by an increase in the entropy. We tried to find the correlation between the entropy contribution to the Gibbs energy of the sublimation process of compounds under study and the structural characteristics of their crystals.

The thermodynamic functions of the sublimation depend on various parameters, including the structure and topology of the molecules, the crystal packing, the molecular conformation, the hydrogen bond network, etc. The *β*-parameter is a reverse characteristic of the molecular packing density in the crystal, which shows how much the free molecular volume in the crystal lattice changes with an increase in the van der Waals volume. Obviously, the variation in the free volume is not additive due to complex correlations between a topology and conformational mobility. The calculated values of the *β*-parameter are given in Table 3. It is apparent that a slight decrease in the free volume of NFT is due to the presence of the hydrogen bond, which leads to an increase in the molecular packing density in the crystal lattice. Moreover, the optimization of the isolated molecules showed that the NFT molecule becomes more planar in a gas phase (Appendix A). The angle between the planes of the furanyl and imidazole rings for NFT decreases from 5.25° to 0.02°. Whereas the FZL molecules both in the crystal and in gas phase have practically unchanged conformation. This behavior of the NFT molecule can lead to an increase in the entropic contribution to the Gibbs energy of the sublimation.

As we have shown earlier [26], there is a correlation between the standard values of the sublimation Gibbs energies and the melting points of the structurally related compounds (selected by the Tanimoto index). A similar approach was used for the studied compounds. The structurally related compounds for NFT and FZL with 0.7 < *T_c_* < 1 were selected using our database, which includes the sublimation thermodynamic functions of the molecular crystals [27]. Moreover, only those compounds were selected that had the values of both Gibbs energy and enthalpy of sublimation, as well as the melting points. The dependence of the experimental sublimation Gibbs energy on the melting temperatures for the compounds structurally related to NFT and FZL (Appendix A) is presented in Figure 6a. Evidently, there is a correlation between the analyzed characteristics, which can be described by the equation:
(1)$$\Delta_{cr}^{g}G_{m}^{0}(298.15\text{K})/\text{kJ}\cdot {\text{mol}}^{-1}=-(70\pm 22)+(0.272\pm 0.046)\cdot T_{fus}/\text{K}$$
R = 0.9342; σ = 6.60 kJ⋅mol^−1^; n = 7

In order to fully describe the sublimation of the compounds under study, it is necessary to predict the sublimation enthalpies. For these purposes, we used a linear relationship between the sublimation enthalpies and Gibbs energies (the so-called “compensation effect”). The analysis results are shown in Figure 6b and can be described by the correlation equation:(2)$$\Delta_{cr}^{g}H_{m}^{0}(298.15K)/\mathrm{kJ}\cdot {\mathrm{mol}}^{-1}=(33\pm 12)+(1.38\pm 0.19)\cdot \Delta_{cr}^{g}G_{m}^{0}(298.15K)/\mathrm{kJ}\cdot {\mathrm{mol}}^{-1}$$
R = 0.9545; σ = 7.97 kJ⋅mol^−1^; n = 7

Thus, the applied clusterization algorithm of the thermodynamic database makes it possible to evaluate all sublimation thermodynamic characteristics (Gibbs energy, enthalpy and entropy) using the derived correlation equations and knowing only one experimental characteristic of the compound—the melting temperature. The presented approach is a convenient and inexpensive way to evaluate the sublimation thermodynamics of the cluster with the structurally related compounds, considering that the melting point is obtained as a result of routine DSC experiments, which is a quick and low-cost method.

### 2.4. Solubility Study

The solubility data of NFT, NFT monohydrate and FZL in the studied solvents at temperatures ranging from 298.15 K to 313.15 K are listed in Table 4 and graphically shown in Appendix A.

Solubility implies the equilibrium between solid phase and solution. It was previously found that NFT spontaneously transforms into the monohydrate form ([NFT+H_2_O]) during its dissolution in water [28]. Preliminary solubility experiments in buffers pH 2.0 and 7.4 at 298.15 K have shown that NFT is unstable and undergoes a solution-mediated transformation from anhydrate form to monohydrate (*Pbca* form) in the solid phase. It should be noted that the conversion occurs in less than 1 h time, since no reflections of anhydrate are detected by the PXRD analysis of the residual materials (Appendix A).

The obtained values of the NFT solubility in water with different pH values are in good agreement with the literature data [29]. Since this source provides equilibrium solubility data for NFT, we compared it with the values we obtained for the monohydrate form. In addition to the data on the solubility of [NFT + H_2_O] in buffer solutions, the literature contains data on the solubility of NFT in 1-octanol [30], with which our experimental data also agree well.

To estimate the thermodynamic characteristics of the NFT dissolution, we obtained the apparent solubility by the analyzing the change in the concentration of the compound at different durations of the experiment and temperature in buffer solutions pH 2.0 and 7.4 (Figure 7). The experiment showed that the rate of the transition of anhydrate NFT to [NFT + H_2_O] depends on the acidity of the dissolution medium. The dissolution profiles show that the solubility decrease in buffer solution with pH 2.0 occurs much more slowly than that in buffer solution with pH 7.4. This is associated with the difference in the solubility of the compound in the dissolution media.

It is seen that the dissolution profile of anhydrate NFT (*β*-polymorph) demonstrates “spring and parachute” behavior. The value of the maximum NFT concentration was used to calculate the thermodynamic functions of the anhydrate form dissolution, while for hydrate NFT the concentration value after 24 h of the experiment was used.

Analysis of the effect of pH on the proportion of the ionized and non-ionized species of APIs showed that FZL is in a non-ionized state over the entire range of the acidity of solutions of interest, whereas for NFT, the proportion of the ionized species increases with an increase in the pH value of the dissolution medium (Appendix A). The experimental data confirm that FZL does not change its ionization state with a change in pH, therefore, the values of the dissolution thermodynamic parameters in buffer solutions pH 2.0 and 7.4 are identical. The dissolution curves and thermodynamic functions of this process differ markedly for NFT. The solubility of NFT in buffer pH 7.4 is increased as a result of its ionization.

However, the level of the solubility for the compounds under study in the selected media is comparable. The solubility of NFT both in buffer solutions and in 1-octanol is higher than that of FZL over the entire temperature range. The transition of anhydrous NFT to [NFT + H_2_O] leads to a decrease in solubility, but a stronger effect of the temperature is observed for the monohydrate form. If at 298.15 K, the equilibrium solubility of NFT monohydrate is more than two times lower than the solubility of anhydrous NFT, then at 313.15 K, this difference does not exceed 15%.

Since the solubility of the compounds under study in the buffer solutions and 1-octanol are quite low, the activity coefficients in these solvents were assumed to be equal to 1. The thermodynamic functions of the dissolution for NFT (anhydrate and monohydrate forms) and FZL and the solvation processes for anhydrous NFT and FZL in buffers (pH 2.0 and 7.4) and 1-octanol are presented in Table 5. The solvation of 1 mol of solute molecules in the solvent can be defined as the total change of the standard thermodynamic functions (Δ*G*, Δ*H*, Δ*S*) of the compound when transferring it from the gas phase (ideal gas; single molecules without interaction) into the solvent. The thermodynamic functions of the solvation process were calculated on the basis of the solubility and sublimation experimental data according to the following equation:(3)$$\Delta Y_{solv}=\Delta Y_{sol}-\Delta Y_{sub}$$
where *Y*—thermodynamic function (*G*—Gibbs energy, *H*—enthalpy, *S*—entropy) of solvation (solv), dissolution (sol) and sublimation (sub) processes.

As follows from Table 5, all dissolution processes in buffer solutions and in 1-octanol are endothermic. This indicates that the enthalpies of solvation do not outweigh the corresponding energies of the crystal lattice. Enthalpy and entropy terms ($\Delta \varsigma_{H_{solv}}$,$\Delta \varsigma_{TS_{solv}}$), which are used to describe the relative fraction of the solvation enthalpy and entropy, show that for the compounds under study, all solvation processes are enthalpy-determined.

PXRD and DSC analysis of the residual phase after the NFT dissolution experiments reveals that the most stable phase in aqueous solutions is the NFT monohydrate (*Pbca* form) (Appendix A), which coincides with the reported data [31]. In the crystal structure of this form, water molecules form robust dimers with NFT by two bifurcated H-bonds with N and O atoms. In addition, the water oxygen atom accepts a moderate N−H···O bond and a C−H···O bond from adjacent NFT molecules (Appendix A). Together, these interactions contribute total 97 kJ·mol^−1^ to the lattice energy of [NFT + H_2_O] (Appendix A). This value is significantly higher than the computed lattice energy of ice I (59 kJ·mol^−1^ [32]), indicating the remarkably high thermodynamic stability of the hydrate. Indeed, the dehydration temperature of [NFT + H_2_O] (Appendix A) is higher than the boiling point of pure water. Such a high dehydration temperature indicates that a large amount of the free energy is required to remove water molecules from the hydrate.

From the DSC and TG experiments, it is possible to obtain the vaporization enthalpy of water from the [NFT + H_2_O] crystal (Δ*H_w_*) according to the following relationship:(4)$$\Delta H_{w}=(\Delta H_{desolv}^{T}\cdot 100/\Delta m)\cdot M_{w}$$

The resulting Δ*H_w_* equals 60.3 kJ·mol^−1^, which is higher than the vaporization enthalpy of pure water (39.2 kJ·mol^−1^ [33]). While it is not possible to perform the sublimation experiment for [NFT+H_2_O], the sublimation enthalpy of the monohydrate form can be estimated from a thermodynamic cycle as the difference between the sublimation enthalpies of [NFT+H_2_O] and anhydrous NFT:(5)$$\Delta_{cr}^{g}H_{m}^{0,T}([NFT+H_{2}O])=\Delta_{cr}^{g}H_{m}^{0,T}(NFT)+\Delta H_{w}$$

The sublimation enthalpy value of NFT monohydrate computed using Equation (5) equals to 200.2 kJ·mol^−1^, which nearly equals to the theoretical lattice energy of [NFT + H_2_O] at the desolvation temperature obtained according to Equation (13) (203.1 kJ·mol^−1^). Hence, the proposed scheme allows the reliable estimation of the sublimation enthalpies of the crystalline hydrates of organic compounds based on thermophysical data and sublimation enthalpies of pure compounds.

It is also worth mentioning that the obtained values are in an almost perfect agreement with the activation energy of dehydration process reported in the work by Koradia et al. (196–197 kJ·mol^−1^) [34]. One may suggest then, that the desolvation process of [NFT+H_2_O] involves the complete destruction of the crystal lattice of the hydrate with the immediate crystallization of the anhydrous NFT (*β*-form).

In contrast to NFT, a similarly packed FZL was not found to form stable hydrates even after stirring the suspension for 72 h. It is not unexpected, since the H-bond with the absent N−H fragment contributes more than a third of the *E_bind_* value. Placing the water molecule into the same crystalline environment surrounded by three FZL molecules leads to the formation of only four intermolecular contacts instead of six with the total energy equal to 51 kJ mol^−1^ (Appendix A). Thus, the formation of a hypothetical FZL hydrate would be energetically unfavorable.

## 3. Experimental Section

### 3.1. Materials

Information about the chemicals used in this study is listed in Table 6. All the chemicals were used without further purification. The sample of NFT was identified as monoclinic *β*-polymorph [17] via powder X-ray diffraction analysis (Appendix A).

### 3.2. Powder X-Ray Diffraction (PXRD)

The powder X-ray diffraction data of the solid samples were collected at room temperature with a D2 PHASER XRD diffractometer (Bruker, Germany) using Cu-Kα1 radiation (λ = 1.54187 Å) and operating at 30 kV and 10 mA. The data were collected from 5 to 30° 2θ with a step size of 0.02° and a count time of a least 1 sec per step.

### 3.3. Differential Scanning Calorimetry (DSC)

The thermal behavior of NFT and FZL, including the melting temperature, was studied with a PerkinElmer Pyris 1 differential scanning calorimeter (Perkin Elmer Analytical Instruments, Norwalk, CT, USA) with Pyris software for Windows NT. The DSC runs were performed in an atmosphere of flowing (20 mL·min^−1^) dry helium gas of high purity (99.996%) in standard aluminum sample pans at a heating rate of 10 K·min^−1^. The DSC was calibrated with an indium sample from Perkin-Elmer (P/N 0319-0033). The melting temperature for indium was 429.75 K (determined by at least ten measurements). The value measured for the fusion enthalpy corresponded to 28.69 J·g^−1^ (reference value 28.66 J·g^−1^ [35]). The DSC measurements were repeated three times.

### 3.4. DSC/TG/Mass-Spectrometry

The thermal stability of the compounds was studied on a NETZSCH STA 409 CD/7/G + Skimmer DSC/DTA/TG with a Skimmer mass-spectrometric vapor analysis system (E_ion_ = 70 eV) in an argon flow at a rate of 70 mL∙min^−1^. A powder sample was heated in a platinum crucible at a rate of 10 K∙min^−1^. The data were collected over the 50–250 m/z (mass) range. The Pt10%—Pt-Rh thermocouple was used for temperature measurements. The mass-spectrometer was calibrated with eight high-purity substances: biphenyl (99.5%), KNO_3_ (99.999%), RbNO_3_ (99.99%), KClO_4_ (≥99%), Ag_2_SO_4_ (99.999%), CsCl (99.999%), K_2_CrO_4_ (≥99%) and BaCO_3_ (99.98%).

### 3.5. Vapor Pressure Measurements

The vapor pressures of NFT and FZL were measured by the transpiration method. Details of the technique are given in the literature [36]. In brief: a flow of the inert gas (N_2_) through the tube transported the saturated vapor of the sample under investigation at a constant temperature and flow rate. The sample vapor is completely condensed at some point downstream. The sample vapor pressure at this temperature was calculated from the sublimated sample amount and the volume of the inert gas used. The mass of sublimated substance was determined using a spectrophotometer method by means of spectrophotometer Cary-50 (Varian, Australia). The stability of the gas flow was achieved by using an MKS 2179A mass flow controller.

The equipment was calibrated with benzoic acid, as specified in earlier work of our research group [37]. The equipment calibration by the carrier gas flow rate showed that the optimal range of the gas flow rate was from 1.2 to 1.8 dm·h^−1^. At this flow rate, the saturated vapor pressure is independent of the flow rate and, thus, the thermodynamic equilibrium is reached. The saturated vapor pressure values were measured five times at each temperature with the standard deviation of no more than 5%. As the saturated vapor pressure of the investigated compounds was low, it could be assumed that the vapor heat capacity change with the temperature was so small that it could be neglected.

Absolute vapor pressures for compounds are usually approximated by the Clarke–Glew equation [38]. Due to the fact that the experimental temperature range at which the saturated vapor pressures FZL and NFT were measured does not exceed 30 K, changes in heat capacity can be neglected and the experimentally determined vapor pressure data may be described in the following way:(6)$$\ln(p/Pa)=A+\frac{B}{T}$$

The value of the sublimation enthalpy is calculated by the Clausius–Clayperon equation:(7)$$\Delta_{cr}^{g}H_{m}^{0}(T)=-R\cdot \left(\frac{\partial (\ln p)}{\partial (1/T)}\right)$$

The sublimation entropy at a given temperature *T* is calculated by the following relation:(8)$$\Delta_{cr}^{g}S_{m}^{0}(T)=\frac{\left(\Delta_{cr}^{g}H_{m}^{0}(T)-\Delta_{cr}^{g}G_{m}^{0}(T)\right)}{T}$$
with $\Delta_{cr}^{g}G_{m}^{0}(T)=-RT\ln(p/p_{0})$, where *p_0_* is the standard pressure (1 × 10^5^ Pa).

For experimental reasons, the sublimation data are obtained at high temperatures. However, in comparison with the effusion methods, the temperatures are much lower, which makes extrapolation to room conditions easier. To further improve the extrapolation to room conditions, we estimated the heat capacities ($\Delta_{cr}^{g}C_{p}(298.15K)$-value) of the crystals using the additive scheme proposed by Chickos et al. [25]. Heat capacity was introduced as a correction for recalculating the sublimation enthalpy $\Delta_{cr}^{g}H_{m}^{0}(T)$-value at 298 K ($\Delta_{cr}^{g}H_{m}^{0}(298.15K)$-value), according to the equation [25]:(9)$$\Delta_{cr}^{g}H_{m}^{o}(298.15K)=\Delta_{cr}^{g}H_{m}^{0}(T)+\Delta_{cr}^{g}C_{p}(T-298.15K)$$

### 3.6. Single Crystal X-ray Diffraction (SCXRD)

The single crystal X-ray diffraction data were collected using a Bruker SMART APEX II diffractometer with graphite-monochromated MoKα radiation (λ = 0.71073 Å) at 298 K. Adsorption corrections based on measurements of equivalent reflections were applied [39]. The structure of FZL was solved by direct methods and refined by full matrix least-squares on *F*^2^ with anisotropic thermal parameters for all the non-hydrogen atoms [40]. All the hydrogen atoms were found from a difference Fourier map and refined with isotropic thermal parameters. Crystallographic data of FZL are given in Table 1. The crystallographic data were deposited with the Cambridge Crystallographic Data Centre as supplementary publications under the CCDC number 2079637. This information may be obtained free of charge from the Cambridge Crystallographic Data Centre at https://www.ccdc.cam.ac.uk/structures/ using the doi of the article.

### 3.7. Solubility Experiment

#### 3.7.1. Equilibrium Solubility Determination

All the solubility experiments were carried out using the analytical isothermal shake-flask method. The study was performed at four temperature points (T = 298.15, 303.15, 308.15 and 313.15 K) and constant pressure (*p* = 0.1 MPa). In brief, an excess amount of API was added to triplicate Pyrex glass tubes containing the respective dissolution media. The tubes were then transferred to an air thermostat supplied by a stirring device for 72 h. After 72 h, the supernatants were filtered through a 0.22 μm PTFE syringe filter (Rotilabo) and diluted with the corresponding solvent to the required concentration. The concentrations of the dissolved APIs were measured spectrophotometrically using a Cary 50 UV-vis spectrophotometer (Varian, Australia) with an accuracy of 2–4%. The wavelength corresponding to the absorption maximums of the studied APIs in dissolution media have been specified as 265 nm in buffer pH 2.0 and 274 nm in buffer pH 7.4 for NFT, 260 nm in buffer pH 2.0 and pH 7.4 for FZL, and 272 nm in 1-octanol for both NFT and FZL.

Conversion of molarity to mole fraction concentration scale was produced using Equation (10):(10)$$X_{2}=\frac{M_{2}S_{2}}{S_{2}(M_{2}-M_{1})+1000\rho }$$
where *S*_2_ is molarity of API (mol·L^−1^), *M*_1_ and *M*_2_ are the molar masses of solvent and solute, respectively, and *ρ* (g·cm^−3^) is the density of the pure solvents. The mole fraction solubilities for buffer solutions were calculated taking into account the buffer compositions. Densities of buffer solutions (pH 2.0 and 7.4) were measured and published early [41]. The used values of densities are given in Appendix A.

#### 3.7.2. Dissolution Study

The dissolution profiles of the anhydrate NFT were carried out by the shake-flask method in buffers pH 2.0 and 7.4 at four temperature points (T = 298.15, 303.15, 308.15 and 313.15 K) and constant pressure (*p* = 0.1 MPa). The excess amount of NFT was suspended in 5 mL of the buffer solution in Pyrex glass tubes. Aliquots of the suspension were withdrawn at predetermined intervals, filtered through a 0.22 μm PTFE syringe filter (Rotilabo), and the concentration of NFT was determined with a suitable dilution by a Cary 50 UV−vis spectrophotometer (Varian, Australia) at the reference wavelength. The results are stated as the average of at least three replicated experiments. The NFT anhydrate stability during the dissolution experiments was monitored by analyzing the solid phase of samples after 1, 5, 10, 30 and 60 min using PXRD.

### 3.8. Computation Details

#### 3.8.1. Solid-State DFT Calculations and Energy of Intermolecular Interactions

Solid-state DFT computations were performed using CRYSTAL17 software [42] at the B3LYP-D3/6-31G(d,p) level of theory. D3 dispersion correction with Becke–Johnson dampening proposed by Grimme [43,44] was used both in structure optimization and in wave function calculation for Bader analysis. The space groups and unit cell parameters of the considered two-component crystals obtained in the single crystal X-ray study were fixed, and structural relaxations were limited to the positional parameters of atoms. As the starting point in the solid-state DFT computations, the coordinates of heavy atoms were used directly from experiments with hydrogen atom positions normalized to the standard X‒H distances from neutron diffraction data. The default CRYSTAL options were used for the level of accuracy in evaluating the Coulomb and Hartree–Fock exchange series and a grid used in evaluating the DFT exchange-correlation contribution. Tolerance on energy controlling the self-consistent field convergence for geometry optimizations and frequencies computations was set to 10^−10^ Hartree. The mixing coefficient of Hartree–Fock/Kohn–Sham matrices was set to 25%. The shrinking factor of the reciprocal space net was set to 4. Frequencies of normal modes were calculated within the harmonic approximation by numerical differentiation of the analytical gradient of the potential energy with respect to atomic position [45]. Quantum topology analysis was performed in Topond14 [46] software currently built into CRYSTAL suit. The (3, −1) critical points search was performed using a standard algorithm and the following quantities were computed in bond critical point (BCP): electron density, *ρ_b_*, its Laplacian, ∇^2^*ρ_b_*, and the positively defined local electronic kinetic energy *G_b_*. The energy of a particular non-covalent interaction, *E*_int_, is computed using the following equation [47]:*E*_int_ (kJ·mol^−1^) = 1124·*G_b_* (atomic units)(11)

The lattice energy *E*_latt_ was estimated as a sum of energies of non-covalent interactions in an asymmetric unit [48]:(12)$$E_{\mathrm{latt}}(0\;K)\;(\mathrm{kJ}\cdot {\mathrm{mol}}^{-1})=\sum_{i}\sum_{j<i}E_{\mathrm{int},j,i}$$
where *j* and *i* denote the atoms belonging to different molecules. Equation (12) is free of basis set superposition error (BSSE). For the sake of simplicity, indexes *j* and *i* will be omitted below.

A thermal correction equal to 2RT [48] was added to compare the theoretical values with the sublimation enthalpy of a crystal at 298.15 K derived from the transpiration experiment:*E*_latt_(T) = *E*_latt_(0 K) + 2RT(13)

In an alternative scheme, the sublimation enthalpy was estimated from vibrational thermodynamics of the crystal and isolated molecules according to our previous study [49].

#### 3.8.2. Van der Waals Volumes Calculation

Van der Waals molecular volumes ($V^{vdw}$) were estimated by the program GEPOL [50] using atom radiuses proposed by Kitaigorodskii [51]. The free molecular volume in the crystal lattice was calculated on the basis of the single crystal X-ray diffraction data:(14)$$V^{free}=V^{mol}-V^{vdw}$$
where $V^{mol}$ is the molecular volume in crystal lattice, which is estimated by the equation:(15)$$V^{mol}=V^{cell}/Z$$
where $V^{cell}$ is the unit cell volume and *Z* is the number of molecules in the unit cell.

#### 3.8.3. Tanimoto Similarity Indices (Clusterization)

The structural similarity was estimated using Tanimoto similarity indices (*T_c_*) obtained in the MOLDIVS (MOLecular DIVersity & Similarity) program [52,53].
*T_c_* = *N*(*A&B*)/[*N*(*A*) + *N*(*B*) − *N*(*A&B*)](16)
where *N*(*A*) is the number of fragments in molecule A, *N*(*B*) is the number of fragments in molecule B, *N*(*A&B*) is the number of common fragments in molecules A and B.

In this program, molecular fragments are defined as atom-centered concentric environments. The fragments consist of a central atom and neighboring atoms connected to it within a predefined sphere size (the number of bonds between the central and edge atoms). For each atom in a fragment, the information on the atom and bond type, charge, valency, cycle type and size is coded into fixed-length variables, which are subsequently used to determine the pseudo-random hash value for this fragment. The program estimates the similarity of every molecule in the database with all the other molecules sorting them by affinity with the initial molecule.

## 4. Conclusions

The single crystal of furazolidone was obtained by slow solvent evaporation, and the crystal structure was determined from the single crystal X-ray diffraction. It was found that the furazolidone molecules are characterized by the layered packing, while the arrangement of the hydrogen-bond chain of nitrofurantoin molecules is of the slipped stacking type. The analysis of the non-covalent interaction energies revealed that the total energy of the contacts between the parallel layers in the furazolidone crystal is 10 kJ·mol^−1^ higher than that in the nitrofurantoin crystal. The π···π stacking interactions and C−H···O bonds formed by the symmetrically located C2−H2 and C3−H3 groups and oxygen atoms of the molecules located in the adjacent layers were found to play an important role in the stabilization of the furazolidone layers. Hirshfeld surface analysis has confirmed that the additional C−H···O interactions in the furazolidone crystal only partially compensate the absence of the stronger N−H···O bond.

Thermophysical studies of nitrofurantoin and furazolidone solids have been carried out using experimental and theoretical methods. Based on the results of the mass-spectrometric analysis, the main degradation products of the studied compounds during melting were identified. The saturated vapor pressures of the compounds were measured in the temperature range (423.1÷467.4) K using the transpiration method. Based on the experimental data, the standard sublimation thermodynamic functions were calculated. The theoretical lattice energy values at 298.15 K agree well with the experimental sublimation enthalpies of both compounds: 146.5 ± 0.7 vs. 142.1 kJ·mol^−1^ for nitrofurantoin and 127.9 ± 1.1 vs. 124.4 kJ·mol^−1^ for furazolidone. By using the space clusterization approach, the correlation between the Gibbs energy, melting temperatures and sublimation enthalpy of the structurally related compounds has been found.

The solvation thermodynamic functions of the studied compounds in buffer solutions with pH 2.0, 7.4 and in 1-octanol were calculated based on the solubility and sublimation experimental data. Due to the tendency of nitrofurantoin to transform into the monohydrate form during the dissolution in water, the thermodynamic parameters of the dissolution process of nitrofurantoin were calculated from the values of the apparent solubility. It was proven that for the compounds under study, all solvation processes are enthalpy-determined. It was found that the rate of the transition of anhydrate nitrofurantoin to monohydrate form depends on the acidity of the dissolution medium. The thermodynamic stability of nitrofurantoin monohydrate was accessed. A scheme for the estimating of the sublimation enthalpy of the crystalline hydrates of the organic compounds based on the thermophysical data and the sublimation enthalpies of pure compounds is proposed. Basing on the experimental, computational and literature data, a mechanism of the desolvation of nitrofurantoin monohydrate upon heating consisting of the complete degradation of the crystal lattice with the subsequent recrystallization of the anhydrous nitrofurantoin (β-polymorph) was proposed.

## Figures and Tables

**Figure 1 molecules-26-03444-f001:**
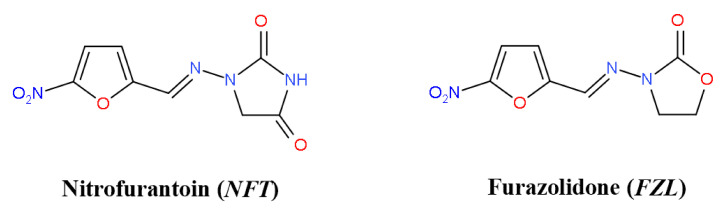
Chemical structures of nitrofurantoin and furazolidone.

**Figure 2 molecules-26-03444-f002:**
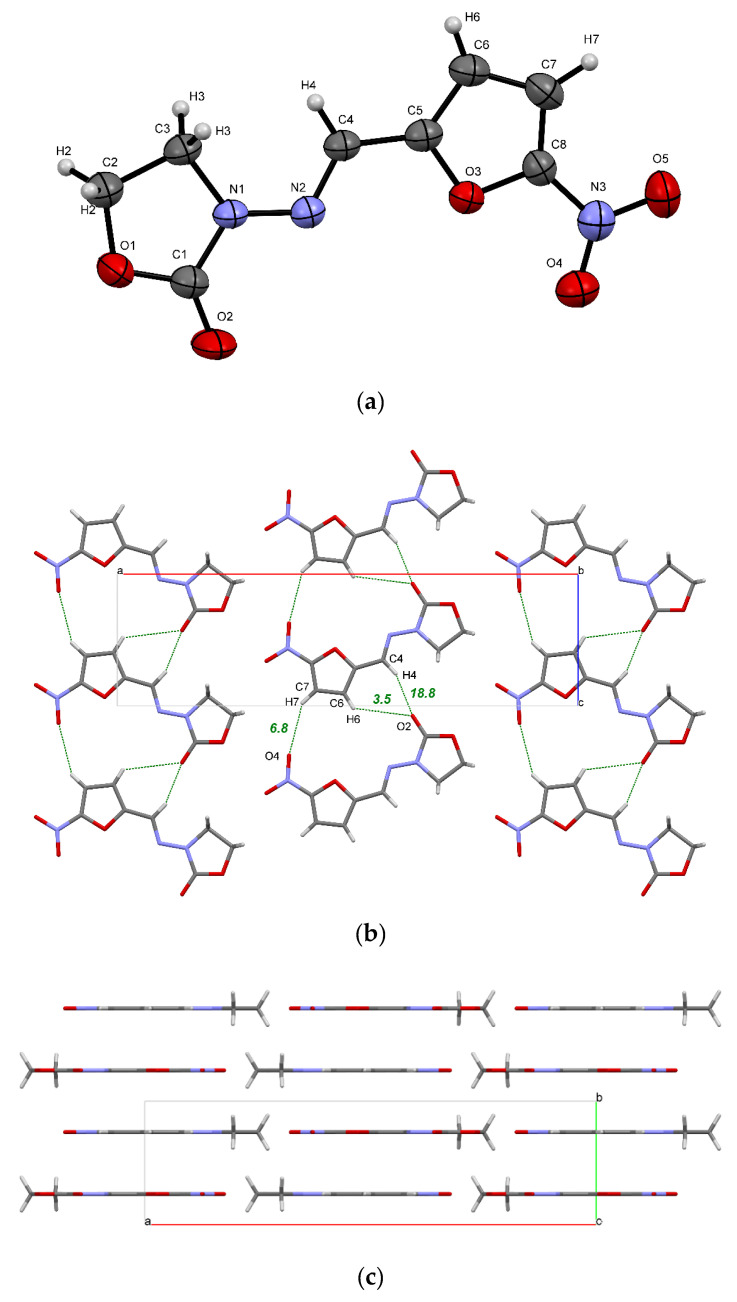
(**a**) ORTEP illustration of an asymmetric unit of the FZL crystal, showing the atom-labelling scheme. Thermal ellipsoids are drawn at the 50% probability level; (**b**) a 2D layer of FZL molecules formed by C−H···O bonds (green dots) and weak interactions; (**c**) packing of 2D layers in the crystal. The numbers indicate the energy of H-bonds estimated using Equation (11) in kJ·mol^−1^.

**Figure 3 molecules-26-03444-f003:**
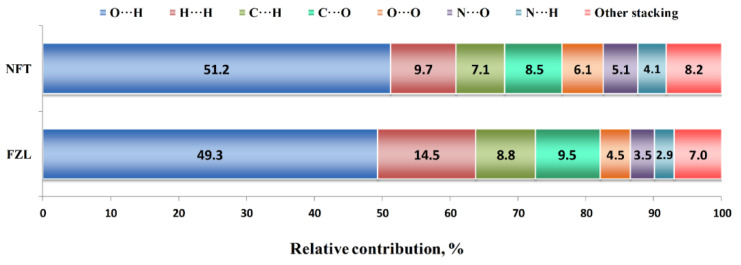
Results of the Hirshfeld surface analysis of NFT and FZL crystals.

**Figure 4 molecules-26-03444-f004:**
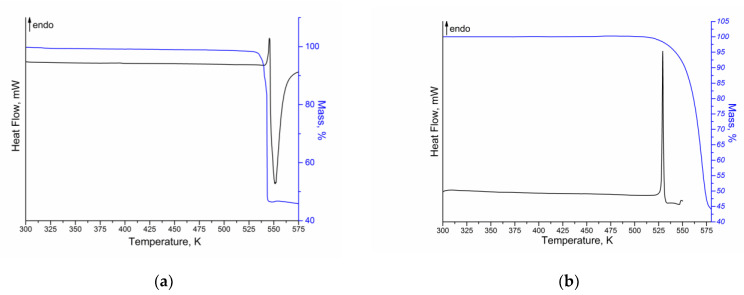
DSC (black) and TG (blue) curves of (**a**) NFT and (**b**) FZL.

**Figure 5 molecules-26-03444-f005:**
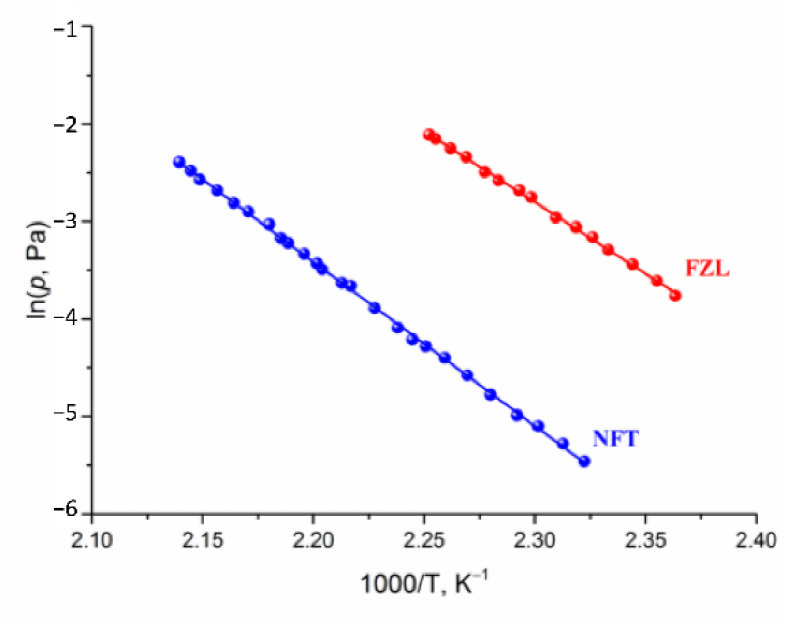
Plots of vapor pressure against reciprocal temperature of the compounds studied.

**Figure 6 molecules-26-03444-f006:**
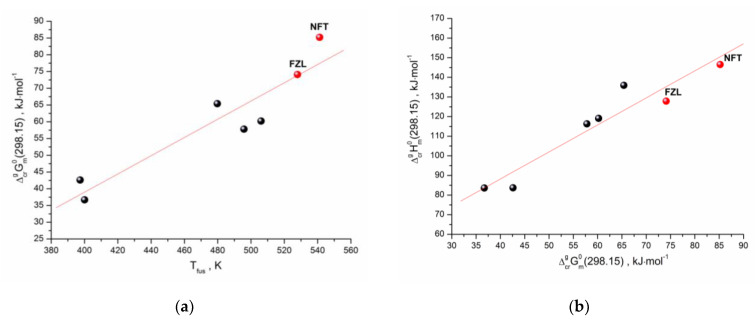
(**a**) Correlation between melting temperature and sublimation Gibbs energy; (**b**) Gibbs energy and enthalpy of sublimation for the set of compounds. The red circles represent results for the studied compounds.

**Figure 7 molecules-26-03444-f007:**
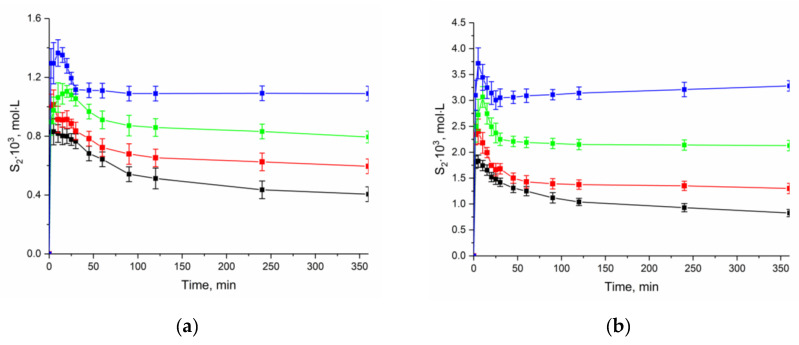
Effect of temperature on the dissolution profiles of the anhydrate NFT in buffer solutions (**a**) pH 2.0 and (**b**) pH 7.4. (–■– 298.15 K; –■– 303.15 K; –■– 308.15 K; –■– 313.15 K).

**Table 1 molecules-26-03444-t001:** Crystallography parameters of FZL.

Empirical Formula	C_8_H_7_N_3_O_5_
Formula weight	225.17
Crystal system	Orthorhombic
Space group	*P*nma
*a* (Å)	22.7977(13)
*b* (Å)	6.2422(3)
*c* (Å)	6.5085(5)
Unit cell volume (Å^3^)	926.21(10)
*Z*	4
Calc. density (g·cm^−3^)	1.615
Absorption coefficient (mm^−1^)	0.137
*θ* range (deg)	1.79–26.00
No. of measured, independent (*R*_int_) and observed reflections	3737, 988 (0.069), 816
Final *R* indices [*I >* 2*σ(I)*]	*R*_1_ = 0.0416, *wR*_2_ = 0.1024
Final *R* indices (all data)	*R*_1_ = 0.0519, *wR*_2_ = 0.1079
Goodness of fit	1.039
max/min Δρ (e·Å^−3^)	0.188/−0.245

**Table 2 molecules-26-03444-t002:** Contributions of different types of stabilizing non-covalent interactions into the lattice energy of NFT and FZL in kJ·mol^−1^ and % of the *E*_latt_(0K) value (in brackets).

	NFT	FZL
*E*_latt_(0K)	139.6	121.9
Molecules in chain	33.8 (24.2%)	28.5 (23.4%)
Chains in layer	40.7 (29.2%)	18.6 (15.3%)
Layers in crystal	65.2 (46.7%)	74.8 (61.3%)
*E* (N−H···O)	27.9 (20.0%)	0 (0%)
Σ*E* (C−H···X)	52.9 (37.9%)	55.4 (45.4%)
Σ*E* (π···π)	58.8 (42.1%)	60.0 (49.2%)
Σ*E* (H···H)	0 (0%)	6.6 (5.4%)

**Table 3 molecules-26-03444-t003:** Sublimation thermodynamic parameters of the investigated compounds.

	NFT	FZL
*p*(298.15K), Pa	1.19·10^−10^	1.12·10^−8^
$\Delta_{cr}^{g}G_{m}^{0}(298.15)$, kJ·mol^−1^	85.2	74.1
$\Delta_{cr}^{g}H_{m}^{0}(T)$, kJ·mol^−1^	139.9 ± 0.7	122.0 ± 1.1
^a^$C_{p}^{cr}(298.15K)$, J·mol^−1^·K^−1^	287.8	286.4
$\Delta_{cr}^{g}H_{m}^{0}(298.15K)$, kJ·mol^−1^	146.5 ± 0.7	127.9 ± 1.1
$\Delta_{cr}^{g}S_{m}^{0}(298.15K)$, J·mol^−1^·K^−1^	205.5 ± 2.8	180.6 ± 4.6
$298.15\cdot \Delta_{cr}^{g}S_{m}^{0}(298.15K)$, kJ·mol^−1^	61.3	53.9
^b^$\varsigma_{H}$, %	70.5	70.4
^b^$\varsigma_{TS}$, %	29.5	29.6
^c^*β*, %	36.3	37.5

^a^ $C_{p}^{cr}(298.15K)$ was calculated according to Chickos’ additive scheme [25]. ^b^ $\varsigma_{H}=(\Delta_{cr}^{g}H_{m}^{0}(298.15K)/(\Delta_{cr}^{g}H_{m}^{0}(298.15K)+T\cdot \Delta_{cr}^{g}S_{m}^{0}(298.15K))\cdot 100\%$. $\varsigma_{TS}=(T\cdot \Delta_{cr}^{g}S_{m}^{0}(298.15K))/(\Delta_{cr}^{g}H_{m}^{0}(298.15K)+T\cdot \Delta_{cr}^{g}S_{m}^{0}(298.15K))\cdot 100\%$. ^c^
*β* = *V*^*free*^/*V*^*vdw*^.

**Table 4 molecules-26-03444-t004:** Temperature dependences of solubility (*X_2_*, mol. frac. and *S_2_*, mol·L^−1^) of NFT and FZL in buffer solutions (pH 2.0 and 7.4), 1-octanol at pressure *p* = 0.1 MPa.

Compound	T, K	Buffer pH 2.0 *X_2_*·10^5^ (*S_2_*·10^3^)	Buffer pH 7.4 *X_2_*·10^5^ (*S_2_*·10^3^)	1-octanol *X_2_*·10^5^ (*S_2_*·10^4^)
NFT	298.15	1.49 (0.83)	3.28 (1.82)	3.61 (2.30)
	303.15	1.82 (1.01)	4.33 (2.40)	4.97 (3.17)
	308.15	2.26 (1.25)	5.51(3.06)	6.53 (4.16)
	313.15	2.46 (1.37)	6.72(3.73)	8.26 (5.26)
[NFT + H_2_O]	298.15	0.73 (0.40)	1.49 (0.83)	-
	303.15	1.07 (0.59)	2.34 (1.30)	-
	308.15	1.43 (0.79)	3.83 (2.13)	-
	313.15	1.96 (1.08)	5.90 (3.28)	-
FZL	298.15	3.35 (1.86)	3.31 (1.84)	1.13 (0.72)
	303.15	4.28 (2.38)	4.27 (2.37)	1.70 (1.09)
	308.15	5.40 (3.00)	5.43 (3.02)	2.12 (1.59)
	313.15	7.62 (4.23)	7.65 (4.25)	2.76 (2.24)

**Table 5 molecules-26-03444-t005:** Thermodynamic functions of solubility and solvation processes of NFT and FZL in solvents studied at 298.15 K.

Solvent	*X_2_* (mol.frac.)	$\Delta G_{sol}^{0},\;\mathrm{kJ}\cdot {\mathrm{mol}}^{-1}$	$\Delta H_{sol}^{0},\;\mathrm{kJ}\cdot {\mathrm{mol}}^{-1}$	$T\Delta S_{sol}^{0},\;\mathrm{kJ}\cdot {\mathrm{mol}}^{-1}$	$\Delta S_{sol}^{0},\;J\cdot K^{-1}\cdot {\mathrm{mol}}^{-1}$	$\Delta G_{solv}^{0},\;\mathrm{kJ}\cdot {\mathrm{mol}}^{-1}$	$\Delta H_{solv}^{0},\;\mathrm{kJ}\cdot {\mathrm{mol}}^{-1}$	$T\Delta S_{solv}^{0},\;\mathrm{kJ}\cdot {\mathrm{mol}}^{-1}$	$\Delta S_{solv}^{0},\;J\cdot K^{-1}\cdot {\mathrm{mol}}^{-1}$	$\varsigma_{H_{solv}},\;$ %	$\varsigma_{TS_{solv}},\;$ %
**NFT**											
pH 2.0	1.49·10^5^	27.5	26.6 ± 2.8	−0.9	−3.1 ± 3.7	−57.7	−119.9	−62.2	−208.6	65.8	34.2
pH 7.4	3.28·10^5^	25.6	37.5 ± 1.9	11.9	39.9 ± 2.7	−59.6	−109.0	−49.4	−165.6	68.8	31.2
1-octanol	3.61·10^5^	25.4	42.8 ± 1.7	17.5	58.5 ± 2.5	−59.8	−103.7	−43.8	−147.0	70.3	29.7
**[NFT+H_2_O]**											
pH 2.0	0.73·10^5^	29.3	50.5 ± 1.7	21.1	70.9 ± 2.6	-	-	-	-	-	-
pH 7.4	1.98·10^5^	26.8	71.8 ± 1.2	44.3	148.5 ± 2.0	-	-	-	-	-	-
**FZL**											
pH 2.0	3.35·10^6^	31.3	41.8 ± 3.2	10.5	35.4 ± 4.0	−42.8	−86.1	−43.3	−145.2	66.5	33.5
pH 7.4	3.31·10^6^	31.3	42.8 ± 2.8	11.5	38.4 ± 4.0	−42.8	−85.1	−42.4	−142.2	66.7	33.3
1-octanol	1.13·10^5^	27.7	49.4 ± 4.4	21.7	72.6 ± 5.2	−46.4	−78.5	−32.2	−108.0	70.9	29.1
$\varsigma_{H_{solv}}=(\left|\Delta H_{solv}^{0}\right|/(\left|\Delta H_{solv}^{0}\right|+\left|T\cdot \Delta S_{solv}^{0}\right|))\cdot 100\%$ $\varsigma_{TS_{solv}}=(\left|T\cdot \Delta S_{solv}^{0}\right|/(\left|\Delta H_{solv}^{0}\right|+\left|T\cdot \Delta S_{solv}^{0}\right|))\cdot 100\%$											

**Table 6 molecules-26-03444-t006:** List of chemicals used in this study.

Compound	CAS Register No.	Source	Mass Fraction Purity *
Nitrofurantoin	67-20-9	Sigma-Aldrich	≥0.98
Furazolidone	67-45-8	Sigma-Aldrich	≥0.98
1-Octanol	111-87-5	Sigma-Aldrich	≥0.99
Potassium dihydrogen phosphate	7778-77-0	Merck	≥0.99
Disodium hydrogen phosphate dodecahydrate	10039-32-4	Merck	≥0.99
Potassium chloride	7447-40-7	Aldrich	≥0.99
Hydrochloric acid 0.1 mol·dm^−3^ fixanal	7647-01-0	Aldrich	-

* as stated by the supplier.

## Data Availability

The data presented in this study are available in Appendix A.

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
