# Peer review of "A Combined Experimental and Theoretical Study of Nitrofuran Antibiotics: Crystal Structures, DFT Computations, Sublimation and Solution Thermodynamics"

_molecules, 2021, doi:10.3390/molecules26113444_

Round 1

Reviewer 1 Report

The submitted study is of a very high quality. It combines both theoretical and comprehensive experimental analysis on two important APIs. The level of novelty is high too as the new, important and anticipated crystal structure has been solved. I am sure that this study deserves to be published, however I have some issues and comments presented below that I would like the Authors to consider before my final recommendation.

The continuous line numbering is missing. This would facilitate the review process.

Calculations: I would suggest calculation of the phonon density of states in order to get the accurate values of the solid state thermodynamic properties, which than could be used to calculate the sublimation thermodynamic properties. This would also require calculations for isolated molecules as well.

The crystal structure of NFT obtained in this study should be compared with the previously reported ones.

Page 5, what about the diffuse functions?

Page 7, what about LABJON03? And about AZAXEG?

Page 11, this part is unclear for me “two polymorphs of NFT(α-andβ-) are known, the DSC curves of which are identical without any phase transitions,  indicating the transformation of one polymorphic form into another”. First, I really doubt that two polymorphs can have IDENTICAL DSC curve. It would mean that the lattice enthalpies are identical too, which is very unlikely if not impossible. Second, I can only see DSC for one polymorph. Third, how can this tell us about the phase transformation?

Page 11, the mass spectrometry is not used to asses the “chemical stability” of the compounds. I guess what the Authors wanted to write but this should be rewritten.

Table 4, it would be nice if the Authors compare those experimental values with the ones from DFT calculations, using the approach described in one of my previous comments.

Author Response

Referee #1

The submitted study is of a very high quality. It combines both theoretical and comprehensive experimental analysis on two important APIs. The level of novelty is high too as the new, important and anticipated crystal structure has been solved. I am sure that this study deserves to be published, however I have some issues and comments presented below that I would like the Authors to consider before my final recommendation.

The continuous line numbering is missing. This would facilitate the review process.

Comment 1

Calculations: I would suggest calculation of the phonon density of states in order to get the accurate values of the solid state thermodynamic properties, which than could be used to calculate the sublimation thermodynamic properties. This would also require calculations for isolated molecules as well.

Answer

For an adequate computation of thermodynamic properties such as heat capacity or entropy anharmonic corrections to vibrational modes need to be taken into account. For a crystal, phonon dispersion calculations are also required, see, e.g. a principal scheme for computing thermodynamic properties of 3D crystals in CRYSTAL17 proposed by the developers:

(from http://tutorials.crystalsolutions.eu/tutorial.html?td=vibfreq&tf=vibfreq_tut)

“…Note that for a full thermodynamic analysis the phonon dispersion relation (http://tutorials.crystalsolutions.eu/tutorial.html?td=dispersion&tf=dispersion-ht) should be available in order to have the phonon density of states to be used in the thermodynamic relations. The error is larger for the acoustic bands and low-frequency phonons.”

Since phonon dispersion calculation implies the computation of a supercell, which is very time-consuming as well as the anharmonic correction treatment, we were unable to perform the thermodynamic calculations at an appropriate level because of strict time limits (10 days to revise the manuscript). Therefore, we had to use the results of the vibrational analysis performed in harmonic approximation only at the gamma point (without the phonon dispersion analysis) to obtain temperature corrections and entropy values. The obtained integral thermal and pV corrections to Elatt from 0 to 298.15 K at p = 1 atm were found to be 1.0 kJ/mol for NFT and -0.5 kJ/mol for FZL, which differs from the commonly used 2RT correction (approx. 5 kJ/mol). The sublimation entropy was found to be about 20% higher than the experimental values for both crystals, leading to significant differences in Gibbs energy of sublimation. The attempts to achieve better agreement between experimental and theoretical thermodynamic functions of sublimation require additional investigations which are beyond the scope of the current work.

Comment 2

The crystal structure of NFT obtained in this study should be compared with the previously reported ones.

Answer

In this work, the only FZL crystal structure was obtained for the first time. The crystal structures of NFT two polymorphs have been obtained and solved earlier and it was not the aim of our work to resolve their structures. Comparison of the NFT powder diffractogram with the calculated diffractograms of its polymorphs showed that it is completely identical to the β-form.

Comment 3

Page 5, what about the diffuse functions?

Answer

We agree that addition of diffuse functions is important for accurate modeling of molecules and molecular clusters, especially when electronegative atoms are present; however, these functions are seldom used in periodic DFT computations due to problems with SCF convergence. From our experience, the addition of diffuse orbitals on multiple atoms often leads to pseudo-linear dependence of the basis set. Unfortunately, very few basis sets with diffuse functions optimized for molecular crystals exist in the literature. Switching to plane wave basis sets is a common option, but lacks the uniform description of periodic (crystal) and non-periodic systems (isolated molecule). Therefore, in our works we use diffuse functions only on halogen atoms, which leads to reasonable agreement with experimental data (see e.g. Levina E.O., Chernyshov I.Y., Voronin A.P., Alekseiko L.N., Stash A.I., Vener M.V. Solving the enigma of weak fluorine contacts in the solid state: a periodic DFT study of fluorinated organic crystals. RSC Adv., 2019, 9, 12520-12537).

Comment 4

Page 7, what about LABJON03? And about AZAXEG?

Answer

Thanks for the comment, missing information about LABJON03 and AZAXEG have been added to the main body of the article in part 3.1 (red color).

Comment 5

Page 11, this part is unclear for me “two polymorphs of NFT(α- and β-) are known, the DSC curves of which are identical without any phase transitions,  indicating the transformation of one polymorphic form into another”. First, I really doubt that two polymorphs can have IDENTICAL DSC curve. It would mean that the lattice enthalpies are identical too, which is very unlikely if not impossible. Second, I can only see DSC for one polymorph. Third, how can this tell us about the phase transformation?

Answer

In this work, our goal was not to obtain and study the polymorphs of nitrofurantoin, therefore all the methods and techniques in the article, including thermal analysis, are used only for study the β-polymorph. DSC curves for both polymorphic modifications of the compound are given elsewhere (Caira, M.R.; Pienaar, E.W.; Lötter, A.P. Polymorphism and pseudopolymorphism of the antibacterial nitrofurantoin. Mol. Cryst. Liq. Cryst. Sci. Technol. Sect. A. Mol. Cryst. Liq. Cryst., 1996, 279, 241-264); a link to this article in our work is given. We agree that the term “identity” of curves is not acceptable, but nevertheless they have similar features. In both cases, the DSC curve has a sharp endotherm corresponding to the melting of nitrofurantoin, followed by a large and sharp exothermic effect associated with the decomposition of the substance. There is no phase transition point on any DSC curve of NFT polymorphs, which would indicate the transformation of one polymorphic form into another. The melting points of the α- and β-forms are very close, differing only by 5 K, as are the densities of crystals with a difference of 0.5%. These data suggest that the α-polymorph is only slightly more stable than the β-polymorph.

Comment 6

Page 11, the mass spectrometry is not used to assess the “chemical stability” of the compounds. I guess what the Authors wanted to write but this should be rewritten.

Answer

We agree that the wording is unfortunate. A mass spectrometric study was carried out to analyze the stability of the studied compounds in the gas phase prior to the sublimation experiment, since both compounds are thermally unstable. The text has been corrected (red color).

Comment 7

Table 4, it would be nice if the Authors compare those experimental values with the ones from DFT calculations, using the approach described in one of my previous comments.

Answer

We agree that such comparison needs to be done; at the same time, mixing the experimental and computed values in Table 4 does not seem reasonable to us. Table SX with comparison of computed and experimental thermodynamic functions of NFT and FZL sublimation process has been prepared and inserted into the ESI, and a brief discussion has been added to Section 3.3 (red color).

Reviewer 2 Report

In section 2.2: change "ambient temperature" by "room temperature

In section 2.3: Change "..repeat triplicate..." by "repeat three times"

In section 2.5: change the phrase "...elevated temperature:." to a more suitable one, e.g. "..high temperature..." (?)

In Figure 4:  indicate which curve is DSC and which TG 

Author Response

Referee #2

Comment 1

In section 2.2: change "ambient temperature" by "room temperature

Answer

The text was changed (red color).

Comment 2

In section 2.3: Change "..repeat triplicate..." by "repeat three times"

Answer

The text was corrected (red color).

Comment 3

In section 2.5: change the phrase "...elevated temperature:." to a more suitable one, e.g. "..high temperature..." (?)

Answer

The phrase was changed

Comment 4

In Figure 4: indicate which curve is DSC and which TG

Answer

The figure caption was revised (red color).

Reviewer 3 Report

This is a very good research - a comprehensive study of crystallographic and some thermodynamic properties of nitrofurantoin and furazolidone, and worthy of publication as a whole. In my opinion, only some parts of the thermodynamic section need to be refined or modified. Below are details.

  1. Equation (1), if used to represent vapor pressure of sublimation, assumes the heat capacity change to be zero and then the enthalpy of sublimation independent of temperature. Although it was determined in the temperature ranges of (430-467) and (423-444) K, it was used to calculate properties at 298.15 K. While extrapolating enthalpy, the authors assumed its temperature dependence by the heat capacity change. But it is not consistent with eqn. (1). I understand the difficulties with far extrapolation, but such an inconsistency should nevertheless be mentioned.
  2. At low pressures, the enthalpy of sublimation for the equilibrium process is practically equivalent to the standard enthalpy of sublimation but it is not the case for entropy and the Gibbs energy. Equation (3) gives standard entropy of sublimation, what I conclude from the superscript, but it should be clearly stated. This is the molar entropy change for the sublimation process at the optional temperature (here 298.15 K) and pressure of 1 bar, and for vapor in the ideal gas state. Although such parameters estimated for conditions very distant from the real processes can sometimes be useful, a detailed analysis of the enthalpy and entropic contributions in the Gibbs energy does not make sense (the text below Tab. 4). The latter contribution decreases as the pressure increases due to the negative change in the entropy of vapor, compressed from the equilibrium pressure to 1 bar.
  3. It is not clear what heat capacity was ultimately used to account for the temperature dependence of the enthalpy of sublimation. Heat capacity of a solid? It should be a difference between vapor and solid heat capacities. Usually the delta symbol appears but not in Table 4.  
  4. Equation (4). If the heat capacity change is a difference between the heat capacity of the vapor and that of the solid, the minus on the right should be replaced with a plus.

Other remarks

  1. P. 4. Correct the name in the reference (Chickos).
  2. Eqns. (7) and (8). Insert a space between 0 and K.
  3. P. 11, line 12 from the top. The Russian “i” letter inserted.
  4. P. 13, the last line. What is Tc? It is not explained.
  5. Fig. 6a. This almost linear dependence results from the Trouton’s rule and the Clausius-Clapeyron equation.
  6. The Detherm database confirms an existence of the solubility data for the NFT + {1-octanol or water} Please compare your data with the literature ones.
  7. Table 5. If the hydrate of NFT is stable and the hydration process occurs continuously, how was it possible to determine equilibrium solubility of the anhydric form (the first row of Tab. 5) ?
  8. Eqn. (14). The solvation process should be defined.  
  9. Ref. 19. Correct the first name of the first author (W instead of X).    

Author Response

Referee #3

This is a very good research - a comprehensive study of crystallographic and some thermodynamic properties of nitrofurantoin and furazolidone, and worthy of publication as a whole. In my opinion, only some parts of the thermodynamic section need to be refined or modified. Below are details.

Comment 1

Equation (1), if used to represent vapor pressure of sublimation, assumes the heat capacity change to be zero and then the enthalpy of sublimation independent of temperature. Although it was determined in the temperature ranges of (430-467) and (423-444) K, it was used to calculate properties at 298.15 K. While extrapolating enthalpy, the authors assumed its temperature dependence by the heat capacity change. But it is not consistent with eqn. (1). I understand the difficulties with far extrapolation, but such an inconsistency should nevertheless be mentioned.

Answer

Indeed, absolute vapor pressures for compounds usually are approximated by the Clarke-Glew equation (Clarke E.C.W., Glew D.N. Evaluation of thermodynamic functions from equilibrium constants. Trans. Faraday Soc., 1966, 62, 539-547). But, because the values of the saturated vapor pressures of the compounds under study are extremely small, it is necessary to work in a narrow temperature range, which for both FZL and NFT does not exceed 30°C. In the investigated temperature range, Eq. (1) is admissible for calculating the enthalpy of sublimation, since the contribution from the change in heat capacity is less than the experimental error and the dependence of the saturated vapor pressure on temperature is linear. When extrapolating to more than 100°C, we, of course, take into account the change in heat capacity, which we indicate in the description of the method.

Comment 2

At low pressures, the enthalpy of sublimation for the equilibrium process is practically equivalent to the standard enthalpy of sublimation but it is not the case for entropy and the Gibbs energy. Equation (3) gives standard entropy of sublimation, what I conclude from the superscript, but it should be clearly stated. This is the molar entropy change for the sublimation process at the optional temperature (here 298.15 K) and pressure of 1 bar, and for vapor in the ideal gas state. Although such parameters estimated for conditions very distant from the real processes can sometimes be useful, a detailed analysis of the enthalpy and entropic contributions in the Gibbs energy does not make sense (the text below Tab. 4). The latter contribution decreases as the pressure increases due to the negative change in the entropy of vapor, compressed from the equilibrium pressure to 1 bar.

Answer

I don't quite understand the term "real processes" you are using. The drugs are objects of our investigations, so the "real processes" and the normal conditions of these processes for drugs are the conditions of their storage and use. Since the values of the saturated vapor pressures of the compounds under study are extremely small, we take them as an ideal gas. The discussion of the sublimation parameters under normal conditions (at 298.15 K and atmospheric pressure) makes sense when it comes to medicinal compounds. The obtained values of the sublimation parameters are further used to calculate the solvation processes and can help in explaining the changes in the dissolution parameters.

Comment 3

It is not clear what heat capacity was ultimately used to account for the temperature dependence of the enthalpy of sublimation. Heat capacity of a solid? It should be a difference between vapor and solid heat capacities. Usually the delta symbol appears but not in Table 4.

Answer

The heat capacity  used to estimate the enthalpy is calculated as follows. If the heat capacities of the solid and gas phases are known (Cpcr and Cpg, respectively), then the standard enthalpy of sublimation, , можно отнести can be attributed to the experimentally measured enthalpy of sublimation at temperature T, , using the Kirchhoff equation:

(for equation see attached file, please)                        (1)

This equation can be used to refer the measured sublimation enthalpy to any temperature. The value of T can denote both the discrete temperature of the calorimetric measurement and the average temperature of the experiment carried out in a narrow temperature range. Taking the heat capacities of the phases independent of temperature and integrating equation (1), we obtain:

 (for equation see attached file, please)             (2)

Heat capacities for many crystalline compounds were measured at 298.15 K [E.S. Domalski, E.D. Hearing, J. Phys. Chem. Ref. Data, 1996, 25, 1-548]. However, there are no experimental values of the heat capacities for the gas state of the substances under study and can only be estimated within the framework of statistical mechanics or using the additivity of functional groups method. The use of the additivity of functional groups method is complicated by the fact that additive group values are not known for all substituents. Therefore, the following mathematical technique is most often used: the second term on the right-hand side of equation (2) is replaced by an expression containing one or another temperature coefficient, which is of a universal nature (J. S. Chickos and W. E. Acree Jr. Enthalpies of Sublimation of Organic and Organometallic Compounds. 1910-2001. J. Phys. Chem. Ref. Data, 2002, 2, 537-698). Equations containing such coefficients are especially valuable in cases where additive group values or the value of the crystal heat capacity for the compound under study are absent. However, in other cases, the most appropriate, in our opinion, is the expression containing the value of the heat capacity of the crystalline phase, since, thereby, the features of the molecular structure of the compound under study are taken into account:

 (for equation see attached file, please)                  (3)

where  can be both experimental and calculated value. Equation (3) is valid in the temperature range T = 200 ¸ 500 K. It is the value of  calculated according to the Chikos additive scheme that is presented in Table 4.

This explanation was given in our previous works, which we refer to in this work, so we did not dwell on this in detail.

Comment 4

Equation (4). If the heat capacity change is a difference between the heat capacity of the vapor and that of the solid, the minus on the right should be replaced with a plus.

Answer

Equation (4) was corrected.

Comment 5

  1. 4. Correct the name in the reference (Chickos).

Answer

The name was corrected (red color).

Comment 6

Eqns. (7) and (8). Insert a space between 0 and K.

Answer

Spaces were inserted between 0 and K.

Comment 7

  1. 11, line 12 from the top. The Russian “i” letter inserted.

Answer

The text was corrected (red color).

Comment 8

  1. 13, the last line. What is Tc? It is not explained.

Answer

Tc is a Tanimoto similarity index. The explanation is given in section 2.8.3.

Comment 9

Fig. 6a. This almost linear dependence results from the Trouton’s rule and the Clausius-Clapeyron equation.

Answer

Yes, probably. However, for each group of structurally related compounds, this correlation dependence has its own correlation coefficients, and when working with other structurally similar compounds for which the parameters of the sublimation process are not known, such correlation dependences can be useful for evaluating these values.

Comment 10

The Detherm database confirms an existence of the solubility data for the NFT + {1-octanol or water} please compares your data with the literature ones.

Answer

Thanks for the comment. Comparison with literature data is carried out, part 3.4 was corrected. (red color).

Comment 11

Table 5. If the hydrate of NFT is stable and the hydration process occurs continuously, how was it possible to determine equilibrium solubility of the anhydric form (the first row of Tab. 5)?

Answer

As we mentioned in the section 3.4: «To estimate the thermodynamic characteristics of the NFT dissolution, we obtained the apparent solubility by the analyzing of the change in the concentration of the compound at different duration of the experiment and temperature in buffer solutions pH 2.0 and 7.4. <…> The value of the maximum NFT concentration was used to calculate the thermodynamic functions of the anhydrate form dissolution, while for hydrate NFT the concentration value after 24 h of the experiment was used». To confirm that, the substance is in an unhydrated form at the point where the maximum value of the apparent solubility of nitrofurantoin is reached, the residual materials after the solubility experiment from 1 to 60 minutes of the experiment were analyzed by PXRD. This was done at each temperature of the experiment in both buffer solutions, for nitrofurantoin. The Figure S8, as an example, is presented a comparison of the diffractograms obtained by dissolving nitrofurantoin in a buffer with pH 7.4 at 298.15 K. It can be seen from the dissolution profile of NFT that after 1 minute from the beginning of the experiment, the maximum value of the apparent solubility is reached, and there is an unhydrated substance in the bottom phase.

Comment 12

Eqn. (14). The solvation process should be defined.

Answer

The definition of solvation process was added (red color).

Comment 13

Ref. 19. Correct the first name of the first author (W instead of X).

Answer

The reference was corrected (red color).

Round 2

Reviewer 1 Report

The Authors have made the necessary corrections, following my instructions. I have no further questions.